# CRAFT-MD: A Conversational Evaluation Framework for Comprehensive Assessment of Clinical LLMs

**Shreya Johri**[1][*][†]**, Jaehwan Jeong**[1,4][*]**, Benjamin A. Tran, MD**[5]**, Daniel I. Schlessinger, MD**[6]**,
**Shannon Wongvibulsin, MD, PhD**[7]**, Zhuo Ran Cai, MD**[3]**, Roxana Daneshjou, MD, PhD**[2,3][‡]**, Pranav
**Rajpurkar, PhD**[1][‡]**

[1]Department of Biomedical Informatics, Harvard Medical School
[2]Department of Biomedical Data Science, Stanford University
[3]Department of Dermatology, Stanford University
[4]Department of Computer Science, Stanford University
[5]Medstar Georgetown University Hospital/Washington Hospital Center, Department of Dermatology
[6]Department of Dermatology, Northwestern University
[7]Division of Dermatology, David Geffen School of Medicine at the University of California, Los Angeles

## Abstract

The integration of Large Language Models (LLMs) into clinical diagnostics has the potential to transform patient-doctor interactions. However, the readiness of these models for real-world clinical application remains inadequately tested. This paper introduces the **C**onversational **R**easoning **A**ssessment **F**ramework for **T**esting in **Med**icine (CRAFT-MD), a novel approach for evaluating clinical LLMs. Unlike traditional methods that rely on structured medical exams, CRAFT-MD focuses on natural dialogues, using simulated AI agents to interact with LLMs in a controlled, ethical environment. We applied CRAFT-MD to assess the diagnostic capabilities of GPT-4 and GPT-3.5 in the context of skin diseases. Our experiments revealed critical insights into the limitations of current LLMs in terms of clinical conversational reasoning, history taking, and diagnostic accuracy, emphasising the need to evaluate clinical LLMs beyond static exam-questions. The introduction of CRAFT-MD marks a significant advancement in LLM testing, aiming to ensure that these models augment medical practice effectively and ethically.

## Introduction

Doctor-patient conversations enable physicians to uncover key details that guide their clinical decisions. However, the mounting pressure of escalating patient numbers, lack of access to care (Lasser, Himmelstein, and Woolhandler 2006), short consultation times (Irving et al. 2017; Wong, Vincent, and Al-Sharqi 2017), and the expedited adoption of telemedicine due to the COVID-19 pandemic (Shaver 2022) have presented formidable challenges to this conventional model of interaction. As these factors risk compromising the quality of history taking and thereby diagnostic accuracy (Bubeck et al. 2023), there is an urgent need for innovative solutions that can enhance the efficacy of these crucial conversations.

New advances in Large Language Models (LLMs), could present a potential solution to this problem (Nori et al. 2023; Singhal et al. 2023; Sarraju et al. 2023; Rajpurkar et al. 2022; Lee, Bubeck, and Petro 2023). These AI models have the ability to engage in nuanced and complex conversations, making them ideal candidates for extracting comprehensive patient histories and assisting physicians in generating differential diagnoses (Moor et al. 2023; Ayers et al. 2023; Au Yeung et al. 2023). However, a considerable gap remains in assessing these models' readiness for application in real-world clinical scenarios (Wornow et al. 2023; Shah, Entwistle, and Pfeffer 2023; Ali et al. 2023). The predominant method for evaluating LLMs in the medical field involves medical exam-type questions, with a strong emphasis on multiple-choice formats (Fijačko et al. 2023; Kung et al. 2023; Han et al. 2023). Although there are instances where LLMs are tested on free-response and reasoning tasks (Strong et al. 2023; Nair et al. 2023; Lowell et al. 2001), or for medical conversation summarization and care plan generation (Shanahan, McDonell, and Reynolds 2023), these are less common. However, none of these assessments explore LLMs' ability for engaging in interactive patient conversations, a crucial aspect of their potential role in revolutionizing healthcare delivery.

## Methods

To address the evaluative shortfall, we propose a new framework for evaluation of clinical LLMs, called the **C**onversational **R**easoning **A**ssessment **F**ramework for **T**esting in **Med**icine (CRAFT-MD). CRAFT-MD allows multi-faceted testing of clinical abilities of LLMs, including medical history gathering and open-ended diagnosis, by employing AI agents in simulations to represent patients or graders, rather than relying completely on human evaluators. This strategy significantly enhances the scalability of evaluations and allows for broader and quicker testing, keeping

---

[*]These authors contributed equally.

[†]Correspondence to: sjohri@g.harvard.edu

[‡]These authors share co-senior authorship.

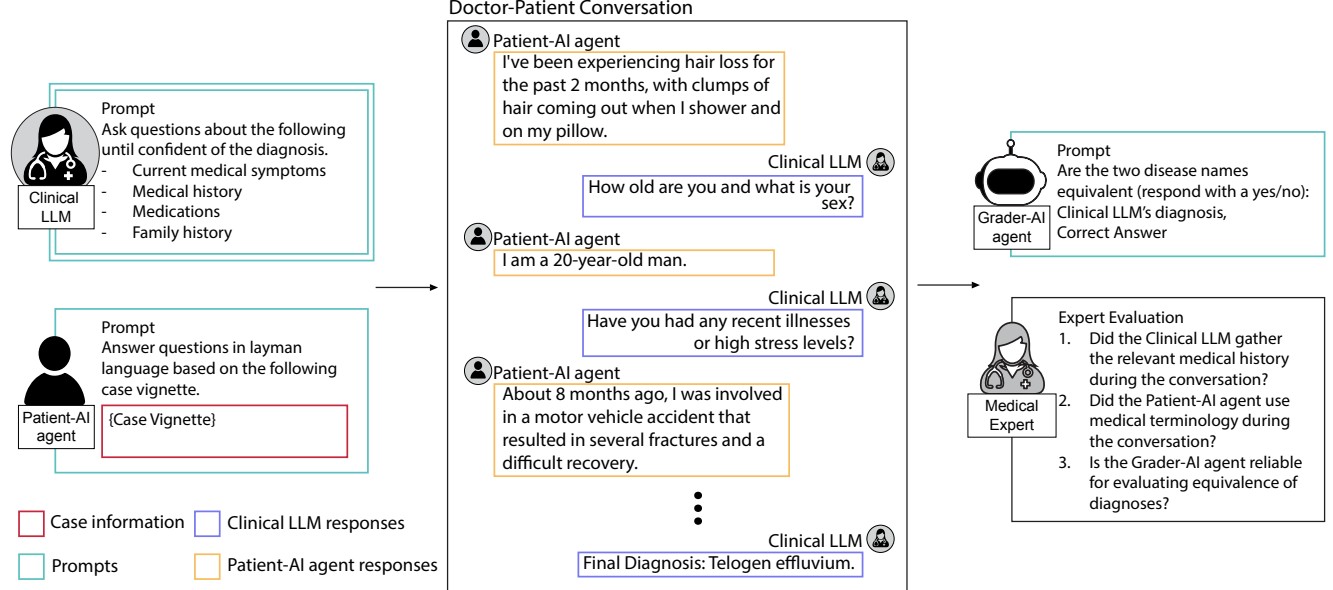

Figure 1: CRAFT-MD evaluates clinical LLMs through simulated doctor-patient consultations with a patient-AI agent using predefined case vignettes. The clinical LLM's objective is to elicit essential medical history from the patient-AI agent and formulate a diagnosis. A grader-AI agent assesses the clinical LLM's accuracy by comparing the clinical LLM's diagnosis to the established ground truth diagnosis. Additionally, medical experts conducts qualitative analysis of the interactions among the clinical LLM, patient-AI agent, and grader-AI agent to thoroughly assess the LLM's clinical reasoning.

pace with the rapid evolution of LLMs (Figure 1).

## Results

We applied the CRAFT-MD framework on 140 case vignettes focused on skin diseases, sourced from both an online question bank[1] (100 cases) and 40 newly created cases, encompassing a variety of skin conditions seen in both primary care and specialist settings. Our evaluations focused on the performance of GPT-4 and GPT-3.5 (versions "gpt-4-0314" and "gpt-3.5-turbo-0301") across 10 simulations per case vignette, revealing several limitations in clinical LLMs' conversational reasoning abilities (Appendix Figure 1, Table 1). In 4-choice multiple choice questions (MCQs), multi-turn conversations decreased accuracy versus vignettes. Notably, multi-turn conversations did not improve over single-turn, but summarizing conversations into concise paragraphs increased accuracy, indicating inability to synthesize across dialogues. Importantly, vignettes had the highest accuracy compared to all conversational setups, indicating limitations in medical history gathering skills. Replacing 4-choice MCQs with free response questions (FRQs), we observed a further decrease in accuracy across all experimental setups, with similar trends for inability to synthesize information and take medical histories. Removing physical exam details further decreased accuracy, indicating potential benefit of multimodal integration in LLMs. Code and data for reproducing experimental results is available online[2].

[1] https://www.clinicaladvisor.com/

[2] https://github.com/rajpurkarlab/craft-md

| Type | GPT-4 | | GPT-3.5 | |
|---|---|---|---|---|
| | MCQ | FRQ | MCQ | FRQ |
| Vignette | 0.919 | 0.684 | 0.833 | 0.546 |
| Multi-turn conversation | 0.854 | 0.431 | 0.724 | 0.468 |
| Single-turn conversation | 0.868 | 0.581 | 0.745 | 0.383 |
| Summarized conversation | 0.856 | 0.607 | 0.810 | 0.474 |
| Multi-turn conversation (without physical exam) | 0.774 | 0.324 | 0.642 | 0.318 |

Table 1: Experimental Results. MCQ = 4-choice Multiple Choice Questions; FRQ = Free Response Questions.

## Conclusion

Recent studies showing high diagnostic accuracy on medical exam questions for LLMs such as GPT-4 may present an overly optimistic outlook for clinical use case, as these evaluations overlook crucial real-world complexities. CRAFT-MD reveals significant deficiencies in LLMs' abilities to gather thorough patient histories, synthesize information over dialogues, and clinical reasoning for diagnosis without answer choices. This work emphasizes the need for responsible and comprehensive evaluation of clinical LLMs.

## Acknowledgments

S.J. is supported by the 2023 Quad Fellowship. We thank the Microsoft Accelerating Foundation Models Research (AFMR) program for providing Azure credits.

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

# Appendix

**a.**
Vignette

Prompt
Based on the symptoms described in the case vignette, select the correct answer choice:

{Case Vignette}

→ Clinical LLM → Diagnosis

{Choice 1, Choice 2, Choice 3 , Choice 4}   OR   No Choices.

**b.**
Multi-turn Conversation

Patient-AI agent

Clinical LLM

Patient-AI agent

Patient-AI agent

Prompt:

{Physical Exam}

→ Clinical LLM → Diagnosis

{Choice 1, Choice 2, Choice 3 , Choice 4}   OR   No Choices.

**c.**
Single-turn Conversation

Patient-AI agent

Prompt:

{Physical Exam}

→ Clinical LLM → Diagnosis

{Choice 1, Choice 2, Choice 3 , Choice 4}   OR   No Choices.

**d.**
Summarized Conversation

Patient-AI agent

Clinical LLM

Patient-AI agent

LLM

Prompt:

{Physical Exam}

→ Clinical LLM → Diagnosis

{Choice 1, Choice 2, Choice 3 , Choice 4}   OR   No Choices.

Case information   Prompts   Clinical LLM responses   Patient-AI agent responses

**e.**

4-option MCQs

GPT-4
- Vignette: 0.919
- Multi-turn conversation: 0.854
- Single-turn conversation: 0.868
- Summarized conversation: 0.856
- Multi-turn conversation (without PE): 0.774

GPT-3.5
- Vignette: 0.833
- Multi-turn conversation: 0.724
- Single-turn conversation: 0.745
- Summarized conversation: 0.810
- Multi-turn conversation (without PE): 0.642

Accuracy

**f.**

FRQs (Single Possible Answer)

GPT-4
- Vignette: 0.684
- Multi-turn conversation: 0.431
- Single-turn conversation: 0.581
- Summarized conversation: 0.607
- Multi-turn conversation (without PE): 0.334

GPT-3.5
- Vignette: 0.546
- Multi-turn conversation: 0.468
- Single-turn conversation: 0.383
- Summarized conversation: 0.474
- Multi-turn conversation (without PE): 0.320

Accuracy

Figure Appendix 1: Schematic showing experimental setups for assessing GPT-4 and GPT-3.5 using CRAFT-MD using (a) vignettes, (b) multi-turn conversations, (c) single-turn conversations, and (d) summarized conversations. (e, f) Clinical LLM's accuracy for all experimental setups on the 140 case vignettes.