# OpenReview forum: "CRAFT-MD: A Conversational Evaluation Framework for Comprehensive Assessment of Clinical LLMs"
_AAAI.org/2024/Spring_Symposium_Series/Clinical_FMs — AAAI 2024 SSS on Clinical FMs_

### Official Review · Reviewer_pUtP · 2024-02-14
**CRAFT-MD: A Conversational Evaluation Framework for Comprehensive Assessment of Clinical LLMs**

**Rating:** 8
**Confidence:** 4

**Review:**

This work proposed an interesting framework for evaluating clinical LLMs. I personally would feel this is a more relevant evaluation framework for real world use cases than traditional benchmarks such as MedQA. The study demonstrated promising capabilities of the framework and revealed limitation of current LLMs in clinical reasoning (at least in the setting without domain-specific in-context learning). However, understandably due to the context length limitation of the non-traditional track, several important technical details are unclear to me. For example, what are the foundation models used in Grader-AI agent, and what's the outcome of expert evaluation?

---

### Official Review · Reviewer_QBwG · 2024-02-19
**An interesting novel approach to automated evaluation of clinical LLMs. However, human validation of the underlying method is required before scaling the method further.**

**Rating:** 6
**Confidence:** 3

**Review:**

1. Summary and contributions: Briefly summarize the paper and its contributions
Introduces a new automated evaluation framework for assessing the conversational ability of LLMs to diagnose medical conditions. The framework uses a patient AI agent (also an LLM) to simulate a doctor-patient interaction. The conclusion of the interaction (the diagnosis) is then evaluated by both a grader LLM and a medical expert. 4 conversational setups are tested using 140 skin disease vignettes. The author claims this shows the shortcomings of LLMs in gathering patient histories, synthesize information over dialogues, and clinical reasoning for diagnosis without answer choices

2. Strengths: Describe the strengths of the work. Typical criteria include: soundness of the claims (theoretical grounding, empirical evaluation), significance and novelty of the contribution, and relevance to the community.
- The development of robust automated evaluation of LLMs is vital for the safe adoption of this technology.
- Increasing work is being done on conversational agents and as is correctly stated in the paper this strategy significantly enhances the scalability of evaluations and allows for broader and quicker testing. Another strength not mentioned over human evaluation is the ease of reproduction.

3. Weaknesses: Explain the limitations of this work along the same axes as above.
- No human baseline is provided and so it is difficult to gauge the level of performance shown in Table 1. It is fair to assume that the drop-off in model performance seen with increasing task complexity, e.g. mcq -> frq, clinical vignette -> multi-turn, would also happen for humans. - The extent of this is unclear though.
- The evaluation dataset is currently limited to 140 cases only focused on skin diseases. Furthermore, an unknown proportion are sourced from the internet. As LLM are trained on large-scale internet crawls, there is a high possibility of test set leakage.

4. Correctness: Are the claims and method correct? Is the empirical methodology correct?
- As an initial paper establishing the framework, this paper needs to first establish the method as valid before making claims establishing the undoubted shortcomings of LLMs. For example, an underlying assumption is that the grader AI agent is accurate and unbiased. Previous LLM evaluation work has shown this not to be true, for example GPT-4 models favouring GPT-4 outputs. A similar assumption holds that the patient AI agent answers accurately and, for example, never hallucinates.
- The validity of the method could be shown by running the same framework swapping out the AI agents for real human and showing the metrics are unchanged. This is suggested by the medical expert box in Fig 1, but no mention of these results is made in the main text.

5. Clarity: Is the paper well written?
Text is clear and well-written
- The icons in Fig 1 make it unclear which parts of the framework are automated, and which are human. For example, the clinical LLM has a human icon, and the grader-AI and patient-AI have different icons.
- It is not clear to me if the patient-ai and grader-ai agents are fixed models (and if so, which models) or change with the clinical LLM model being tested. Both could lead to differing methodological issues. By fixing the models, the framework may favour models of the same “type” but by changing the evaluating models metric is fundamentally changed.

6. Relation to prior work: Is it clearly discussed how this work differs from previous contributions?
- A thorough, succinct review of the need for medical innovation, LLM development and shortcomings in automated medical evaluation is given. The introduction could be explicit about the pros and cons between human and automated evaluation.
- The recent work “Towards Conversational Diagnostic AI” would be highly relevant to mention. But an acceptable miss as it was published very close to the submission deadline.

7. Reproducibility: Are there enough details to reproduce the major results of this work?
- The link to framework’s code is provided for reproducibility.
- It is mentioned that 10 simulations are run. I assume this is done via sampling (e.g. through temperature) but not mentioned what these hyperparameter settings are

---

### Official Review · Reviewer_Hx6p · 2024-02-22
**This paper proposes a method for evaluating conversational reasoning, history taking, and diagnostic accuracy for dermatological conditions, using AI to simulate patients in the conversation. Whilst the evaluation is lacking some important details and the results do not currently support this approach it represents an interesting and important advance in evaluation frameworks and the authors should be commended for attempting this.**

**Rating:** 7
**Confidence:** 4

**Review:**

Thank you for this interesting paper which proposes a method for evaluating conversational reasoning, history taking, and diagnostic accuracy for dermatological conditions, using AI to simulate patients in the conversation.

I find it curious that they chose such a vision-centric specialty to evaluate a text-based model, as this is simply not realistic - no doctor would diagnose a skin condition without looking at it! Nevertheless the authors should be commended for going beyond the ‘simple’ MQA approaches of other works as they describe in the background and I think their agent based solution is promising, despite the performance demonstrated here.

Whilst the focus of this paper is on the evaluation framework the following key details are missing:
The test set vignettes are constructed from an unspecified mixture of dermnet and bespoke cases. The proportion of these should be reported, and performance stratified by case origin, as the dermnet cases will have formed part of the training data for GPT3.5/4
MCQs are used for evaluation but there is no detail on how these were constructed. This is important to confirm whether they were part of model training and to gauge how realistic the questions are - e.g. were the other answers blatantly wrong or plausible differentials. It’s for this latter reason that single best answer (SBA) questions are more commonly used in medical exams nowadays. It’s also unusual for MCQs to have only 4 options, which makes me wonder how they were generated!

I appreciate the word limit but to me the most interesting element of this work is the patient conversation agent and there is little detail on how they did this. For example 10 simulations are mentioned, presumably to capture variation in how patients may present histories for the same disease, but no explanation is given of how this variation is modelled - different prompts, temperature settings etc?

The authors include an appropriate benchmark (the whole ‘vignette’) against both multi-turn conversation, single conversation (unclear why this is useful?) and a summarised version of the vignette. Interestingly the the highest performance is observed when prompting the model with the complete case vignette. Whilst the authors suggest this indicates limitations in medical history gathering skills, however without knowing the case mix I think it is more likely that the model is simply remembering the dermnet vignettes from it’s training. I would imagine that then the summarisation, single and multi-turn conversations are then adding progressively increasing noise/corruption of the original vignette and this is what is hurting performance.

The authors include qualitative ‘expert’ evaluation in their methods but do not show any of the results of these, including whether the grader-AI agent reliably evaluated equivalent diagnoses (a central requirement for their evaluation framework).

Despite these critiques and the results shown, I believe this study is an important demonstrator of an agent-based conversational evaluation framework and is worthy of acceptance.

---

### Official Review · Reviewer_tAc4 · 2024-02-23
**CRAFT-MD advances LLM evaluation for healthcare, but its focus on specific models limits generalizability. A more balanced perspective and comparative analysis with Retrieval-Augmented Generation models would enhance its relevance in the evolving clinical LLM landscape.**

**Rating:** 7
**Confidence:** 5

**Review:**

**Pros:**

- **Advancement in LLM Evaluation:** CRAFT-MD represents an advancement in Large Language Model (LLM) evaluation, moving beyond static exams to capture the complexity of real-world clinical conversations. This is particularly relevant as LLMs are increasingly being explored for healthcare applications.

- **Scalability and Ethical Considerations:** The use of AI agents for simulations offers scalability, ethical considerations, and control over conversation flow, providing a more controlled environment for evaluation.

- **Comprehensive Evaluation:** CRAFT-MD assesses various aspects, including history gathering, information synthesis, and diagnosis under different conditions, offering a comprehensive evaluation framework.

**Cons:**

- **Comparison with RAG-based Models:** Given the evolving landscape of clinical LLMs, it is suggested to explore the applicability and effectiveness of CRAFT-MD when used against models based on Retrieval-Augmented Generation (RAG). Including a comparative analysis with these models would offer valuable insights into the framework's performance relative to different approaches in the field.

- **Limited Generalizability:** The study focuses specifically on  two LLM models, potentially limiting its generalizability. Further exploration across a wider range of LLMs would strengthen the paper's applicability to broader contexts.

- **Need for Balanced Perspective:** While highlighting LLM limitations is valuable, a more balanced perspective could be achieved by also exploring potential strengths and areas for improvement.